# A Comparison of Different Folding Models in Variations of the Map Folding Problem

**Yiyang Jia [1],\* and Jun Mitani [2]**

1   Faculty of Science and Technology, Seikei University, Tokyo 180-8633, Japan
2   Faculty of Engineering, Information and Systems, University of Tsukuba, Tsukuba 305-8577, Japan;
    mitani@cs.tsukuba.ac.jp
\*   Correspondence: s02966@cc.seikei.ac.jp

**Abstract:** In this paper, we compare the performance of three different folding models when they are applied to three different map folding settings. Precisely, the three folding models include the simple folding model, the simple folding–unfolding model, and the general folding model. The different map folding settings are discussed by comparing different folded states, i.e., different overlapping orders on the set of the squares of $1 \times n$ maps, the squares of $m \times n$ maps, and the squares lying on the boundary of $m \times n$ maps. These folding models are abstracts of manual works and robotics. We clarify the relationship between their reachable final folded states under different settings and give proof of all the inclusion relationships between every two of these sets. In addition, there are nine distinct problems with the three folding models applied to three folding settings. We give the optimal linear time solutions to all the unsolved ones: the valid total overlapping order problems of $1 \times n$ maps, $m \times n$ maps, as well as the valid boundary overlapping order problems of $m \times n$ maps with the three different folding models. Our work gives the conclusion of the research field where the folding models and the overlapping orders of map folding are concerned.

**Keywords:** folding models; robotics; overlapping order

## 1. Introduction

Problems in the field of origami are becoming more and more popular because of their applicability in robotics and computational modeling, especially the origamis whose crease patterns (comprising all the folding line segments and their intersections) are relatively simple. Some examples of how they have been applied in robotics manufacturing and moving paths are proposed in [1–4]. In this paper, we mainly focus on the origamis with grid patterns as their crease patterns, which are called *map folding*. We investigate different variations of map folding problems with three folding models. These models are the *general folding*, the *simple folding*, and the *simple folding–unfolding*. The general folding is an abstract of manual works, while the latter two are abstracts of the folding patterns which robots can manage. By comparing the performance of these folding models in different folding problems, our conclusion provides an important reference for applying these folding models to robotics manufacturing and computational modeling.

To introduce the topic of this research, we present the terminologies at first. Precisely, an $m \times n$ map is a rectangle sheet with m rows and n columns composed of $m \times n$ congruent squares. When m equals 1, the map reduces to one dimension and is usually represented by a line segment with segments in unit length representing squares of the map. The edges of the squares not located at the boundary of the map are called *creases*. They together comprise a *grid pattern* as the *crease pattern* of the map. Two squares sharing the same crease are called a pair of *neighbor* squares. A *complete folding sequence* of the map indicates the sequence of folds that fold the map to the shape of a unit square on the plane. Every single square becomes a unique layer in the folded state. In any partly or completely flat-folded state, we say a pair of squares whose surfaces touch each other are *adjacent*. The

overlapping orders refer to the orders given onto the set of all the squares of the map (total orders) or a subset consisting of only the squares aligning on the boundary of the map (*boundary orders*, illustrated as the shadowed squares in Figure 1a. If such an input order corresponds to the order of these squares aligning from bottom to top in a practical folded state of the map, then it is said to be *valid*.

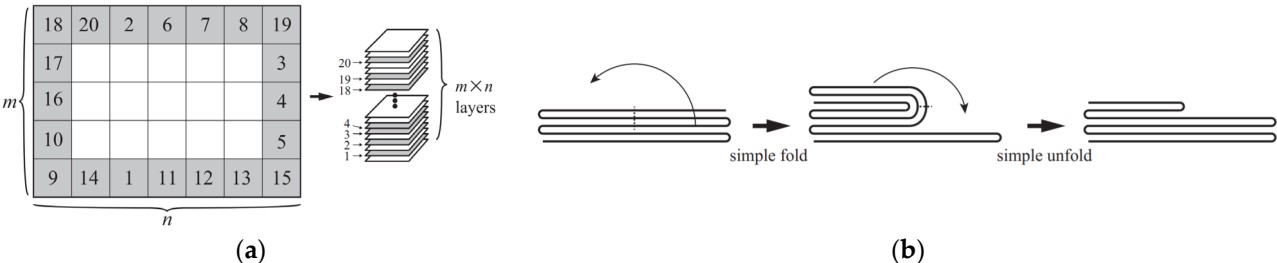

<div align="center">

**(a)** **(b)**

</div>

**Figure 1.** (**a**) A map is folded into $m \times n$ square layers. The overlapping order of the boundary squares in the folded state is given. (**b**) An example of simple fold and unfold.

In this paper, the capability of three folding models: the *general folding*, the *(some-layers) simple folding*, and the *(some-layers) simple folding–unfolding*, are compared in three variations of the map folding problem, which are referred to as *valid overlapping order problems* in this paper: *valid total overlapping order problem of 1 × n maps*, *valid total overlapping order problem of m × n maps*, and *valid boundary overlapping order problem of m × n maps*. For conciseness, in the following, these valid overlapping order problems are referred to as *1D VOP*, *Total VOP*, and *Boundary VOP*, respectively. Their corresponding decision problems concern the decisions on the validity of input overlapping orders.

The general folding model uses general folds with only the practical restrictions of the paper itself, namely, no tearing, stretching, or penetrating (self-intersection). Instead, the simple folding model uses only simple folds while the simple folding–unfolding model comprises both simple folds and its reverses operation, simple unfolds. Both simple folds and unfolds are always applied simultaneously on some top or bottom layers along a single line (crease line) and satisfying that the next state is also flat, as illustrated in Figure 1b. Every simple fold and unfold can be indicated by a set of creases lines.

We will show that although the three folding models have the same reachability of the overlapping order in 1D VOP, their reachable valid overlapping orders in both Total VOP and Boundary VOP are in strict inclusion relations. A conclusion of our results is given in Table 1, where A, B, C with different indexes indicate the sets of reachable overlapping orders. Except for the proved equivalence on the performance between the general folding model and the simple folding–unfolding model in 1D VOP [5], all the other results are proposed for the first time. Furthermore, to complete the solutions of all these decision problems we listed, in this paper, we also give two linear-time solutions to two unsolved problems: one to the decision problem of Total VOP with the simple folding model, the other to the decision problem of Boundary VOP with the general folding model.

**Table 1.** The results of three folding models in three overlapping order problems.

| Problems | Simple Folding | | Simple Folding–Unfolding | | General Folding |
|---|---|---|---|---|---|
| 1D VOP | A | = | A | = | A |
| Total VOP | $B_1$ | $\subsetneq$ | $B_2$ | $\subsetneq$ | $B_3$ |
| Boundary VOP | $C_1$ | $\subsetneq$ | $C_2$ | $\subsetneq$ | $C_3$ |

Existing studies have neither investigated the relationship between different folding models nor considered these folding models under different problem settings. However, to both practical applications and theoretical studies, it is necessary to clarify the performances of these folding models, their distinct advantage and disadvantage when putting them

into different settings, especially when applying them to robotic uses, such as the example given in [6]. To make up for the deficiency of existing studies, in this paper, we combine the three different folding models with three different problem settings. Our conclusion provides a comprehensive, all-sided, and detailed analysis of the folding models.

## 2. Preliminaries and Terminology

Map folding concerns the flat folding of special grid patterns. More precisely, the original map folding problem proposed by Jack Edmonds in 1997 asks the computational complexity when deciding the flat-foldability of a regular grid pattern of size $m \times n$ with a *Mountain-Valley assignment* (*MV assignment*). An MV assignment is a constraint description over the folding direction of every crease as either a mountain ("M", denoted by the red solid-line segments) or a valley ("V", denoted by a blue dashed-line segment). As a problem actually not as trivial as it intuitively seems, it remains unsolved for almost 40 years until today. A simple pattern which is locally flat-foldable at every vertex (which means, every small neighborhood of its vertices is foldable) but not globally flat-foldable is illustrated in Figure 2a to exemplify this point.

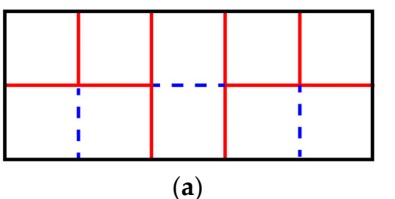
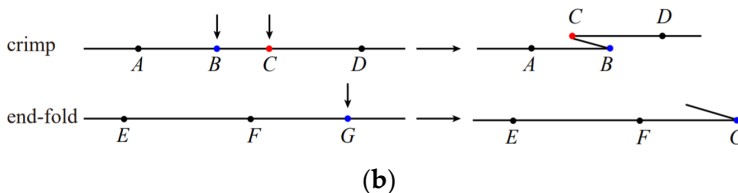

(a)  (b)

**Figure 2.** (**a**) A $2 \times 5$ map which cannot be flat-folded; (**b**) An example of the crimp and the end-fold.

For general patterns, the existence of a flat-folded state and the existence of general folding progress leading to the state is proven to be equivalent [7]. Thus, the flat-foldability, i.e., whether a given crease pattern (with or without a Mountain-Valley assignment) can be flat-folded or not, can be decided through a check on the existence of a valid overlapping of all the faces, which is proven to be NP-hard generally [8]. Although the flat-foldability is globally intractable, locally, the flat-foldability around every single vertex in a small neighborhood involving no other vertex can be decided in linear time by checking some restrictions [9]. Two well-known restrictions for both MV-assigned or unassigned cases were presented as the Kawasaki-Justin Theorem, and the Maekawa-Justin Theorem [10–12].

Arkin et al. first proposed the simple folding model to simplify the map folding problem [13]. Three kinds of simple-folding models, *one-layer simple folding*, *all-layers simple folding*, and *some-layers simple folding*, are considered in their study. Among them, the some-layers simple folding model is the most general model, and we follow it in this paper. As introduced above, every *simple fold* transforms some top or bottom layers along a single line to achieve a state also flat. One-layer simple folding model and all-layers simple folding models specify the number of folded layers to one and the number of all the layers, respectively. In their research, they used two kinds of special simple folds, *crimps*, and *end-folds*, as the basic operations. In a 1D map, as illustrated in Figure 2b, an *end-fold* is a fold at either the first or the last crease point. The interval between the last crease (which becomes a point since the map is illustrated as a line segment) and its corresponding end of the map is no longer than its neighbor interval. A *crimp* is a fold along a pair of adjacent crease points labeled "*MV*" or "*VM*". The length of the interval between the two crease points is a local minimum.

They have probed several kinds of maps, including *one-dimensional maps* with every unit generalized to arbitrary length, *regular maps* as grid patterns of size $m \times n$, and some extended patterns only keeping the orthogonality of the creases. Considering the relationship with this research, we introduce their main results for the first two kinds of maps. They proposed linear-time algorithms to decide the flat-foldability of input MV-assigned 1D maps using one-layer simple folds and some-layer simple folds, as well as a

polynomial-time algorithm for the same problem with all-layers simple folds. All of the results mentioned above were introduced in [13]. Later, Akitaya et al. presented some NP-hard results of these simple folding models for orthogonal patterns, with fixed unit lengths, either with or without MV-assignments [14].

The some-layers simple folding model was further extended by Uehara into the *some-layers simple folding–unfolding model*. This model permits both the some-layers simple folds and its reverse operation, which is actually some-layers simple unfold (referred to as simple folds and simple unfolds in the following) [5]. As mentioned before, it was proven that in a $1 \times n$ map with fixed unit lengths, the valid folded states of the simple folding–unfolding model are exactly the same with the valid folded states of *general-folding model*, and there must exist at least one simple folding–unfolding sequence to reach an arbitrary valid folded state, which contains no more than $2 \times n$ (un)folds. According to the method in [5], it can be further clarified that not only the final flat-folded state but also every reachable partly flat-folded state by general folds can definitely be reached by the simple folding–unfolding model.

To make it clear, both the simple folding–unfolding model and the simple folding model of an $m \times n$ map satisfy Conditions (1) and (2): (1) Index the layers from top to bottom, then each operation is applied to some top or bottom layers with consecutive indexes and is along a single line; (2) The state before or after an operation is always flat and in the shape of a rectangle.

The difference between them is that every crease in the simple folding model can be only folded once and then never unfolded, whereas creases can be either folded or unfolded multiple times in the simple folding–unfolding model. It is evident that a folding following the simple folding model also follows the simple folding–unfolding model, and a folding following the simple folding–unfolding model also follows the general folding model. Namely, the inclusion relation of reachable final flat-folded states in Table 1 is evident. What we aim to prove is the "proper" part in the proper subset relation. The above viewpoint is about the range of final flat-folded states a folding model can reach in different problem contexts. Oppositely, we can also discuss whether a given overlapping order represents a real or unreal folded state of a folding model in a certain problem. That is to say, such a problem concerns the *validity of given overlapping orders* (*VOP*). In [15], VOP considering the total order of $m \times n$ squares in an $m \times n$ map with the general folding model is proven to be linear-time solvable. However, such a result does not give enough hint on the computational complexity of the map folding problem because a flat-foldable map may have exponential flat-folded states.

In [16,17], two linear-time algorithms are proposed to decide: (1) the validity of overlapping orders given on all the squares of an $m \times n$ map (Total VOP) using the simple folding–unfolding model; (2) the validity of boundary overlapping orders of an $m \times n$ map (*Boundary VOP*) using the simple folding model, respectively. In this paper, we will provide the solutions to all the combinations of problem settings and types of models that remained untouched or unsolved. In other words, this work can be seen as a perfection of the evaluation of all these three models in different variations of map folding. Moreover, we will discuss the reachable final flat-folded states sets (the configuration space) of different models in different problems. Despite the same performance of all the models in $1 \times n$ maps (1D VOP), the equality relation over their reachable flat-folded states sets turns into strict inclusion relation as the map becomes two dimensional (i.e., Total VOP). Besides, even if we restrict the overlapping orders to only the boundary squares, the strict inclusion relation still keeps (i.e., Boundary VOP).

## 3. Outline

In Section 4, we will discuss the performance of the three folding models in 1D VOP. Uehara has already proven that every reachable flat-folded state of a $1 \times n$ map with the general folding model is also reachable for the simple folding–unfolding model [5]. This paper gives proof that the remaining one, the simple folding model, performs the same. In

Section 5, we will propose a linear-time solution for making the decision on the validity of total overlapping orders (Total VOP) using the simple folding model. Because of the inclusion relation between the set of boundary squares and the set of all the squares, the strict inclusion relation of the reachable total overlapping orders would become clear once we proved the strict inclusion relation of the reachable boundary overlapping orders. This proof will be given in Section 6. We will also present the way to decide the validity of boundary overlapping orders (Boundary VOP) which uses the general folding model. In Section 7, we provide a discussion on these folding models, their possible applications, their possible extensions, as well as their limitations. We also introduce some references about the applications of these folding models involving both simulation studies and real examples. Finally, in Section 8, we conclude our results and the future work.

## 4. Simple Folding Model in 1D VOP

It has been proven in [5] that both the final flat-folded state and every partly flat-folded state of the general folding model can be reached through a sequence of simple folds and unfolds. In this section, we will prove that although there exist partly flat-folded states of the general folding model unreachable for the simple folding model, their sets of reachable final flat-folded states are always the same.

With the simple folding model, once two layers are folded to touch each other, they would not be separated apart again. This is the difference between the simple folding model and the simple folding–unfolding model on the reachable (partly) folded states. Because of this property, not all the partly flat-folded states of the simple folding–unfolding model can be reached by the simple folding model. An example is given in Figure 3. The red part in Figure 3b is the main reason for the unreachability. The details will be given in the following explanation. However, in the final folded state, all the layers would be aligned over a $1 \times 1$ area (represented by a line segment with 1 in length). In the following, we will show that the folds applied on such red parts can be applied earlier to avoid the simple unfolds. In other words, for any final folded state, we can always find a corresponding simple folding sequence, where every step causes some layers to touch each other.

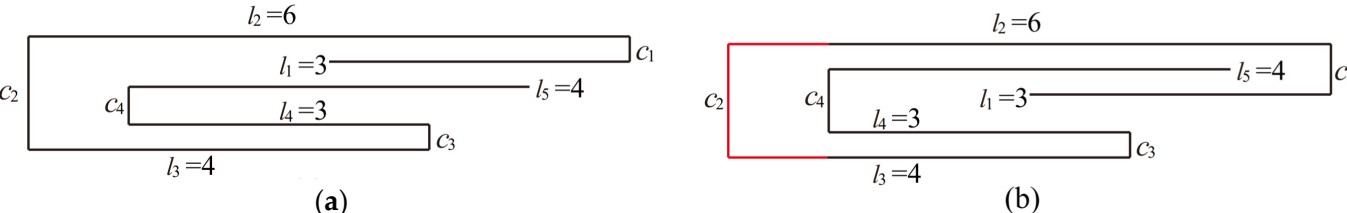

**Figure 3.** A $1 \times 20$ map comprising five segments from $l_1$ to $l_5$ aligning in order. Two input overlapping orders correspond to the states (**a**,**b**). (**a**) is reachable by the simple folding model while (**b**) is not.

We use the same simple folding operation, crimps, and end-folds as [13] to indicate our folding process but neglect the creases remaining unfolded at every step. The two partly folded states after folding a crimp and after folding an end-fold are described in Figure 2b. The size of the map before the fold is assumed to be $1 \times n$ with its left end located on $x = 0$. We denote the coordinates of points $A$ to $G$ as $x_A$ to $x_G$, which are all integers. Then, we have $x_B - x_A > x_C - x_B < x_D - x_C$ for the crimp and $n - x_G \leq x_G - x_F$ for the end-fold. The two states are respectively described by the following formulas, where $o(x)$ indicates the squares folded to the coordinate $x$, $s_i$ ($0 \leq i \leq n$) indicates the squares whose lower-left vertex is located at point $i$ before any fold, and tuples indicate the order of the squares from bottom to top.

For the crimp, we have:

$$o(x) = \begin{cases} s_x, & x < 2x_B - x_C \\ \left(s_x, s_{2x_B - x - 1}, s_{x + 2x_C - 2x_B}\right), & 2x_B - x_C \leq x < x_B \\ s_{x + 2x_C - x_B}, & x_B \leq x < 2x_B + n - 2x_C \end{cases} \tag{1}$$

and for the end-fold, we have

$$o(x) = \begin{cases} s_x, & x < 2x_G - n \\ (s_x, s_{2x_G - x - 1}), & 2x_G - n \leq x < x_G \end{cases} \tag{2}$$

The MV assignment of the map is uniquely determined by the input overlapping order, as described in [15]. The assignment can be computed in time linear to $n$. To make the latter computing easier, we first utilize their approach to determine the MV assignment.

**Step 1**. Compute the MV assignment of the map. Then, repeat the following three steps until either the map is shrunk to size $1 \times 1$ or the input order is determined to be unreachable. In the former case, the complete simple folding sequence could also be obtained.

**Step 2.** Find out the neighbor segments which are adjacent in the overlapping order. Decide the crease between them to be the first crease to fold. If no such neighbor segment exists, the input overlapping order is unreachable.

**Step 3.** According to the creases decided in Step 2, decide the crimps and end-folds on the 1D map by referring to Formulas (1) and (2). If no feasible crimp or end-fold exists, the input order is unreachable.

**Step 4**. Reduce the map to a new (smaller) map by applying the folding operations. Go back to Step 2 until the map is reduced to size $1 \times 1$.

This process can be performed in linear time of the 1D map since it can be finished by a standard graph traverse algorithm, e.g., a breadth-first search or a depth-first search algorithm. Thus, we have the following theorem.

**Theorem 1.** *The simple folding model and the simple folding–unfolding model have the same reachable final flat-folded states.*

**Proof of Theorem 1.** Theorem 1 can be concluded by mathematical induction. To prove Theorem 1, it is sufficient to prove that for every folded state without penetration. We can unfold the map to its initial state using only simple unfolds but no simple folds (a crease unfolded would not be folded again). (1) The conclusion is clear for a $1 \times 1$ map. (2) We assume that for a $1 \times k$ map, the conclusion holds. Since a $1 \times (k + 1)$ map can be obtained by gluing a square at the end of the $1 \times k$ map, we have the cases as illustrated in Figure 4. (a) indicates the case where the $(k + 1)$-th square is adjacent to the $k$-th square. In this case, we only have to consider the $k$-th square and the $(k + 1)$-th square as a whole. The unfolding process can be finished by unfolding all the other creases for the original $1 \times k$ map and finally unfolding the crease between the $k$-th square and the $(k + 1)$-th square. (b) indicates the case that the $k$-th square and the $(k + 1)$-th square are not adjacent and the end of the $(k + 1)$-th square is visible from outside (which is on the opposite of (c). In (c), the end of the $(k + 1)$-th square is invisible), this time the unfolding can be performed by considering the $(k + 1)$-th square together with the layers below it, first unfold the part $p_2$ and then unfold $p_1$ as the supposed unfolding of the original $1 \times n$ map. (c) indicates the case that the $k$-th square and the $(k + 1)$-th square are not adjacent and the end of the $(k + 1)$-th square is invisible from outside. This time, first unfold $p_3$, then unfold the outside (in this figure, the bottom) layer of $p_2$ as end-fold. After these unfolds, $p_2$ and $p_3$ becomes a whole part below the $(k + 1)$-th square. Unfold this whole part as a single layer, then unfold the $(k + 1)$-th square, and finally, unfold the remaining unfolded part of $p_2$ and $p_1$ according to the unfolding process of the original $1 \times n$ map. In all the illustrations, $p_1$, $p_2$, and $p_3$ can be arbitrarily complicated; the other end of the map can also be extended to comprise all the possible cases. However, the unfolding process always follows the above manner. Correspondingly, we proved that for any $1 \times (k + 1)$ map, its reachable folded states of the general folding are the same as its reachable folded states of the simple folding. Theorem 1 is proven. □

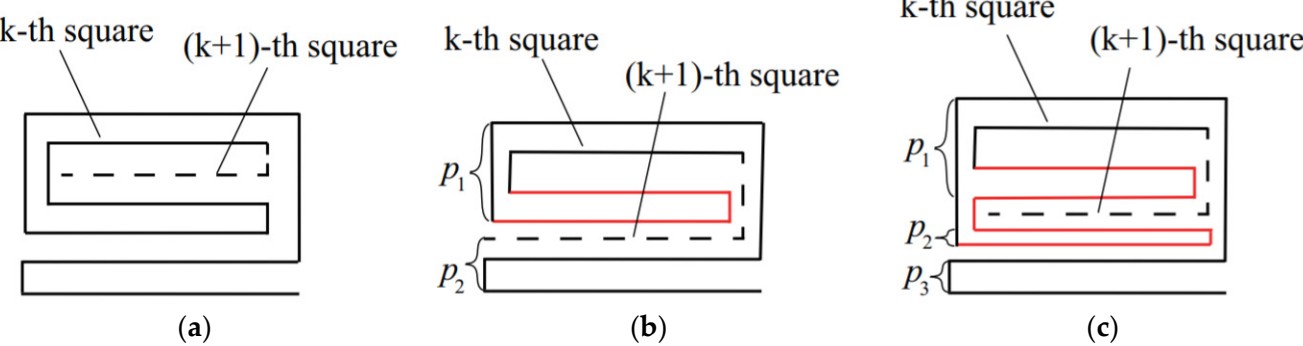

**Figure 4.** Three different cases of when gluing the $(k + 1)$-th square to a $1 \times k$ map. The dashed segments illustrate the $(k + 1)$-th square. (**a**) indicates the case where the $(k + 1)$-th square is adjacent to the $k$-th square; (**b**) indicates the case that the $k$-th square and the $(k + 1)$-th square are not adjacent and the end of the $(k + 1)$-th square is visible from outside; (**c**) indicates the case that the $k$-th square and the $(k + 1)$-th square are not adjacent and the end of the $(k + 1)$-th square is invisible from outside.

## 5. Simple Folding Model in Total VOP

It has been introduced in Section 1 that for an $m \times n$ map, no matter if a single operation follows the simple folding model or the simple folding–unfolding model, the state before and after the operation would always be flat and in the shape of a rectangle. There already exists a method to decide the validity of given total overlapping orders with the simple folding–unfolding model. When the model changes to the simple folding model, it can be solved with a similar method, which maintains most of the existing one while with a little adjustment [16]. The adjustment is, when considering the order of the parallel crease lines supposed to be folded continuously, instead of using the folding–unfolding process introduced in [5], we repeat the reduction by finding the adjacent relation of neighbor squares to decide the next fold at each step. The conclusion is given as Theorem 2.

**Theorem 2.** *The validity of the input total overlapping order with the simple folding model can be decided in O(mn) time.*

**Proof of Theorem 2.** We first introduce the flow of the method, and then give the details on the adjustment. (1) Compute the MV assignment; (2) Decide the time points to change the folding direction to either horizontal or vertical. During each phase between two nearby time points, the map would be folded from a rectangular shape to a smaller rectangular shape with the length of one side fixed; (3) Viewing each partly flat-folded state as a new reduced map, a simple fold on a crease line should influence the overlapping order of all the neighbor pairs incident to this crease line in accordance. Traverse the input order to check for the unity and decide the folded state at every time point in (2); (4) Check the validity of the folded state at every time point. If all these computations are managed with all folded states tested as valid ones until the map is reduced to the size of $1 \times 1$, the input is valid. To apply the above process to the simple-folding model, we have to adjust (4) to check if the folded state at every time point is reachable by only simple folds. Since all the neighbor squares on the two sides of a single crease line (creases on a single line segment) are influenced by the folding of this crease line simultaneously, and their accordance is already checked in (3). We can view any partly flat-folded state as a 1D map. The check then becomes: given an input overlapping order of the segments separated by crease points, does it correspond to a valid folded-state of the simple-folding model? In this problem, all the lengths of the segments are integers. Using the same method as introduced in Section 4, the process will either tell that no available fold exists (invalid) or give the sequence of the simple folds leading to an input folded-state. Since every pair of neighbor squares would be handled only once according to this process, the check of (4) costs linear time in total. Following the above process, the validity of every partly flat-folded state can be checked, and the process totally costs $O(mn)$ time, which is the same as the simple

folding–unfolding model. Because the other parts of the method remain the same, the total solution of the Total VOP with the simple folding model costs $O(mn)$ time. Theorem 2 is proven. □

We give the computation for a $1 \times 20$ map illustrated in Figure 3 as an example. The map is in length 20, and it comprises five segments $l_1$ to $l_5$ in order. Two input overlapping orders $(l_3, l_4, l_5, l_1, l_2)$ and $(l_3, l_4, l_1, l_5, l_2)$ correspond to the two states (a) and (b) in Figure 3, respectively. From the discussion before, their MV assignments are unique. For (a), $\{l_3, l_4\}$, $\{l_4, l_5\}$ and $\{l_1, l_2\}$ are the neighbor pairs adjacent in the overlapping order. Therefore, the corresponding creases $c_1$, $c_3$, and $c_4$ are firstly folded. The map is then reduced to the state illustrated in Figure 5a. On the opposite, for the states in Figure 3b, by the fact that $l_3$, $l_4$ forms a neighbor pair adjacent in the overlapping order while $l_5$ and $l_4$ are not adjacent, $c_3$ is firstly folded while $c_4$ is not. Naturally, $l_5$ and $l_2$ should touch each other. However, the adjacent square pairs represent a state, as shown in Figure 5b, which is not valid.

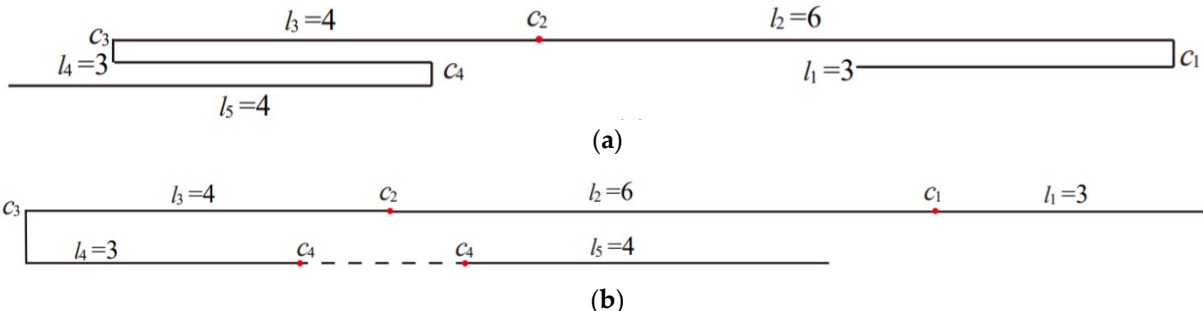

**Figure 5.** The partly folded states of (**a**,**b**) in Figure 3.

## 6. Performances of Three Folding Models in Boundary VOP

### 6.1. General Folding Model in Boundary VOP

The conditions of the general folding model are no stretching, no tearing, and no penetration. The first two are naturally respected as the map being folded to the shape of a single square. Thus, for any given order, no matter a total order or a partial order on the set of all the squares, we only have to check the no penetration condition. There is a basic fact that a pair of horizontal neighbor squares would never penetrate with a pair of vertical neighbor squares, as the three possible overlap states illustrated in Figure 6a.

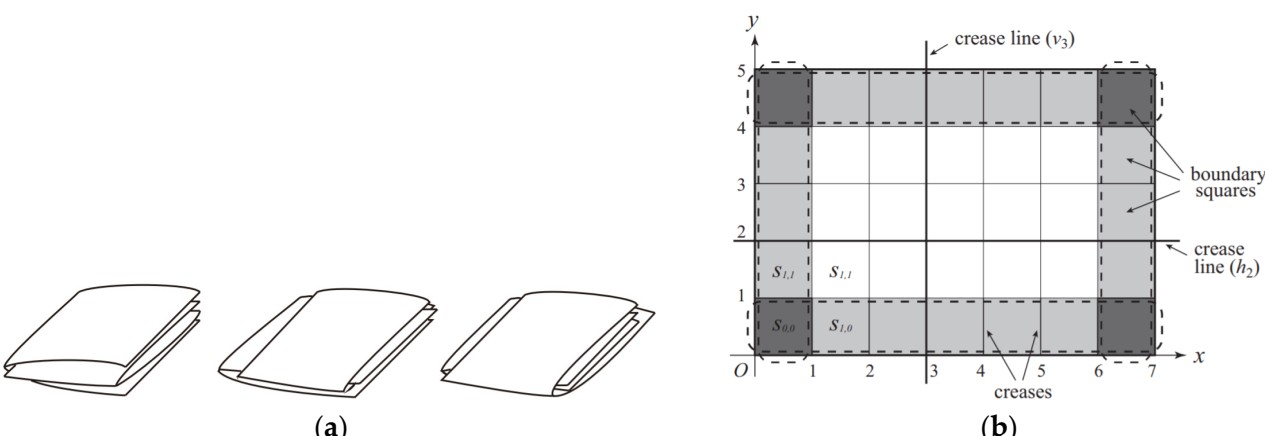

**Figure 6.** (**a**) Three possible overlap states of a pair of horizontal neighbor squares and a pair of vertical neighbor squares. (**b**) The notations and the coordinate system of the map.

Hence, we can consider two horizontal boundary sides and two vertical boundary sides separately. Because of the similarity, here, we only exemplify the computation on two horizontal boundary sides. The following narration follows the notations with the help of the coordinate system given in Figure 6b. $s_{i,j}$ ($0 \leq i < m$, $0 \leq j < n$) refers to the square whose lower-left vertex is located at $(i, j)$ before any fold. We further assume that the upfront side of $s_{0,0}$ is fixed. In the final folded state, on the bottom row, the creases between $s_{2i-1,0}$ and $s_{2i,0}$ would align on the y-axis, the creases between $s_{2i,0}$ and $s_{2i+1,0}$ would align on the line $x = 1$. On the top row, when m is odd (even), the creases between $s_{2i-1,\,m-1}$ and $s_{2i,\,m-1}$ would align on the *y*-axis ($x = 1$), the creases between $s_{2i,\,m-1}$ and $s_{2i+1,\,m-1}$ would align on the line $x = 1$ (*y*-axis).

For every two neighbor square pairs $\{s_{i-1,\,p}, s_{i,\,p}\}$ and $\{s_{j-1,\,q}, s_{j,\,q}\}$ whose creases align on the same line segment when completely folded, whether they penetrate each other or not, should be checked. Any penetrating state can be described as a permutation of ($s_{i-1,\,p}$, $s_{j-1,\,q}, s_{i,\,p}, s_{j,\,q}$). A similar property holds for the two vertical boundary sides. Checking the existence of such orders on four boundary sides totally costs $O(m + n)$ time applying the stack structure introduced in [15].

Conversely, based on the assignment, there exists a relation < on the set of boundary squares, defined as: $s_{c,d} < s_{a,b}$ if $s_{c,\,d}$ is below $s_{a,b}$ in the final state. It is clear that < is a strict partial order because it satisfies:

(**irreflexivity**) there is no $s_{a,b} < s_{a,b}$;
(**asymmetry**) if $s_{a,b} < s_{c,d}$, then there is no $s_{c,d} < s_{a,b}$;
(**transitivity**) if $s_{a,b} < s_{c,d}$ and $s_{c,d} < s_{e,f}$, then $s_{a,b} < s_{e,f}$.

The strict partial order < corresponds to a directed acyclic graph whose nodes are the boundary squares and edges are from $s_{c,d}$ to $s_{a,b}$ when $s_{c,d} < s_{a,b}$. Whether the input order follows a directed acyclic graph or not can be checked by a traverse. To conclude, we have Theorem 3. The proof is omitted because it follows the above analysis. The entire check takes $O(m + n)$ time.

**Theorem 3.** *An input boundary overlapping order is valid if and only if it satisfies the partial order < and includes no penetration. The decision on the validity can be given in O(m + n) time.*

*6.2. Strict Inclusion Relations among Three Folding Models*

As aforementioned, the strict inclusion relation among the folding models in Total VOP follows their strict inclusion relation in Boundary VOP. It is clear that every reachable flat-folded state of the simple folding model is definitely reachable for the simple folding–unfolding model, and every reachable state of the latter one is also reachable for the general folding model. Then, to prove the strict inclusion relation in Boundary VOP, we only have to prove that there do exist reachable boundary overlapping orders of the general folding model that cannot be reached by the simple folding–unfolding model, as well as reachable boundary overlapping order of the simple folding–unfolding model that cannot be accessed by a simple folding model. In the following, we provide concrete instances that respectively belong to the difference sets of their reachable boundary overlapping orders.

For the difference between the general folding model and the simple folding–unfolding model, an instance is given in Figure 7a. The total overlapping order on the left side was first proposed in [7] as an instance that needs complex general folds (not simple folds). Its corresponding boundary overlapping order is given in Figure 7a on the right side. In this instance, since only the center square labeled five in the left figure is not a boundary square, the boundary overlapping order decides the entire MV assignment. Clearly, there exists no crease line entirely assigned to valleys or mountains in this figure, which means that a first simple fold could not be applied. Thus, this instance corresponds to a reachable boundary overlapping order of the general folding model that cannot be reached by the simple folding–unfolding model.

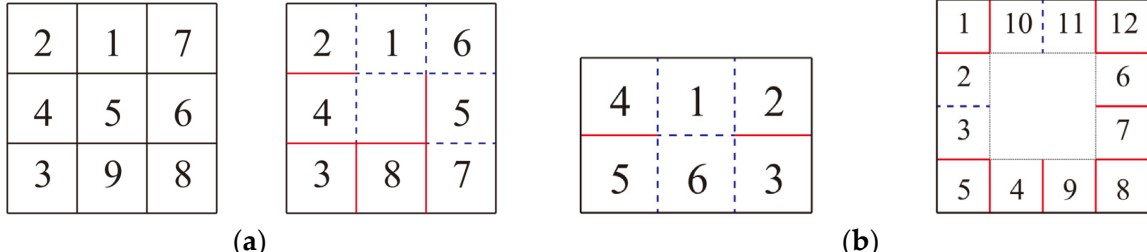

**Figure 7.** (**a**) A boundary overlapping order reachable for the general folding model while unreachable for the simple folding–unfolding model and the simple folding model. (**b**) The left one indicates a boundary overlapping order corresponding to the smallest map with crease lines entirely assigned valleys that is only reachable by the general folding model; The right one indicates another boundary overlapping order corresponding to the map with crease lines entirely assigned valleys that is only reachable by the general folding model.

However, even in an MV-assignment, including crease lines entirely assigned the same, there still exist boundary overlapping orders reachable by the general folding model but unreachable by the simple folding–unfolding model. Such a boundary overlapping order of the smallest map (of size $2 \times 3$) is given on the left side of Figure 7b, where the boundary overlapping order is also the total overlapping order. For the maps with larger $m$ and $n$, the result still holds. An instance is given on the right side of Figure 7b. The above instances naturally bring this question: what kind of boundary overlapping order reachable by the general folding model can also be reached by the simple folding–unfolding model? We give the answer in Theorem 4.

**Theorem 4.** *A valid boundary overlapping order of the general folding model is also valid for the simple folding–unfolding model if and only if every flat-folded state can be separated to rectangles with respect to the order.*

**Proof of Theorem 4.** Considering a valid folding sequence by the simple folding–unfolding model from the last step, the reverse sequence can be described as each time applying all the unfolds and folds along the same direction until no more unfolds along this direction can be applied, then changing the direction and repeating the unfolds and folds. During the process, every fold or unfold is applied on a rectangle shape, and the rectangle would be divided into smaller rectangles by the crease lines. Thus, it is clear that every flat-folded state of the simple folding–unfolding model can be separated into rectangles. Conversely, as long as at each partly folded-state after the direction change can be separated to rectangles before the next time of direction change, the map can be viewed as a $1 \times n$ map where the overlapping of the rectangles can be viewed as the overlapping of line segments. Referencing the result that every partly folded state of a $1 \times n$ map is also reachable by a simple folding–unfolding model, the reverse sequence can be achieved. This ensures that if every flat-folded state can be separated into rectangles, the overlapping by the general folding model must also be valid for the simple folding–unfolding model. □

Next, we give the instance that is valid for the simple folding–unfolding model but is unacceptable for the simple folding model. A map was first folded along the vertical crease lines to a state on the left side of Figure 8a, which, as mentioned before, is achievable by a combination of simple folds and unfolds model but not achievable by only simple folds. In such a folded state, any horizontal crease lines would lock the touching layers, and thus keep them from separating apart again. For easiness, we fold the map along the middle such that two horizontal boundary sides are aligned on the same line segment. As mentioned in Section 1, for the simple-folding model, any subsequent folds will not separate the layers already touched each other. Thus, the two horizontal boundary sides can be viewed as glued together in this partly folded state. The figure on the right side of Figure 8a gives the top view of the folded state. Because a simple folding model can never

reach a state where these touching squares touch each other, the corresponding boundary order is unreachable by the simple folding model.

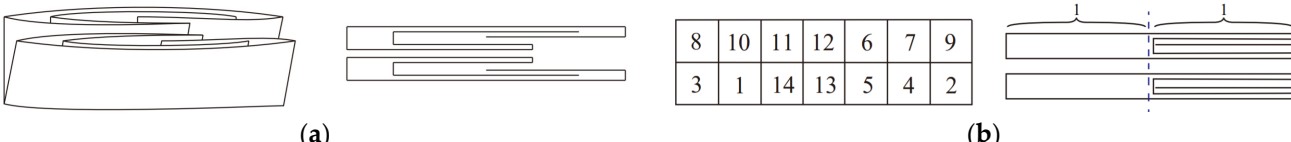

**(a)**                                                        **(b)**

**Figure 8.** (**a**) A boundary overlapping order reachable for the simple folding–unfolding model while unreachable for the simple folding model. (**b**) The simplest instance reachable for the simple folding–unfolding model while unreachable for the simple folding model.

Based on the above idea, we give the most straightforward instance reachable for the simple folding–unfolding model while unreachable for the simple folding model on the left side of Figure 8b, which can be viewed as an instance for both Boundary VOP and Total VOP. The length of each line segment is concreted in the figure on the right side of Figure 8b.

## 7. Discussion

The three folding models: the simple folding model, the simple folding–unfolding model, and the general folding model, have been discussed from the viewpoint of their performances when applying them to different folding problems in the above sections. We compared their capabilities and investigated the relationships between them in different folding problems. Although our analyzing methodology is rather theoretic, these folding models do have practical uses and often show capacities in accord with our analysis in this paper. For example, ref [6] gives the application of a robot built based on the simple folding model in $1 \times n$ maps. Some other simulation studies and analyses on practically manufactured models or robots can be found in [1,3,4,18,19].

The importance of our research is that because we place all the folding models together and then analyzed their capabilities theoretically under different contexts, we can provide a complete, comprehensive, and all-sided result of all the commonly used folding models. However, since this research is rather theoretical than practical, the practical features of these models still remain to be figured out. For the purpose of clarifying their practical performances, this research can be extended using neutrosophic statistics as a future direction [20,21].

## 8. Conclusions and Future Work

In this work, we listed three folding models and compared their capabilities in different map folding problems. Their available valid overlapping orders in $1 \times n$ maps are the same, while their respective sets of valid overlapping total orders and boundary orders in $m \times n$ maps are in strict inclusion relations. For completeness, we also showed that, in $m \times n$ maps, deciding the validity of total overlapping orders with the simple folding model and deciding the validity of boundary overlapping orders with the general folding model could be solved in time linear to the size of the input. Our research makes up for the deficiency of existing results, as no research has ever compared these different folding models or investigated their essential inner associations. In conclusion, our result provides a comprehensive, all-sided, and detailed analysis of these folding models theoretically. Some practical studies on the possible applications of these models are desired to be inspired by this work.

The decision problems concerning other subsets of the squares with the three folding models can be considered as interesting future work. In addition, these models can have extensions in more generalized maps. Although the NP-hardness of the decision on the flat-foldability of given orthogonal crease pattern with simple folds is proven [14], the complexity of deciding the validity of given overlapping orders with different folding models remains unknown.

**Author Contributions:** Conceptualization, Y.J. and J.M.; investigation, Y.J.; methodology, Y.J.; supervision, J.M.; writing—original draft preparation, Y.J.; writing—review and editing, J.M. All authors have read and agreed to the published version of the manuscript.

**Funding:** This research received no external funding.

**Institutional Review Board Statement:** Not applicable.

**Informed Consent Statement:** Not applicable.

**Conflicts of Interest:** The authors declare no conflict of interest.

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
