# Peer review of "A Comparison of Different Folding Models in Variations of the Map Folding Problem"

_applsci, doi:10.3390/app112411856_

Round 1

Reviewer 1 Report

Thank you for allowing me to read this interesting and valuable study. It was my pleasure to give the authors several suggestions to help them improve the manuscript further. The paper discusses that comparing the performance of three different folding models when they are applied to three different map folding settings.

 I found this article interesting and indeed highlighting a knowledge gap of importance. However, I have several remarks before the paper can published as is in Applied sciences Journal

It would be good to first correct the formal error and then look at the paper. Moreover, there is no discussion part in the manuscript and Moreover, conclusion part is too short.

Author Response

Sincerely thank you for your advice. Please see the attachment to see our revision.

Reviewer 2 Report

  1. The abstract should be extended and some main results should be discussed.
  2. The novelty of the paper is not clear. How this study is different from the existing studies. A detailed should be added at the end of the introduction.
  3. The references are not well cited. The references should be uniform.
  4. The performance should be compared with the existing ones using the simulation studies.
  5. The performance should be compared with the existing ones using the real example.
  6. The limitations and applications of the proposed study should be discussed. 
  7. The reference list should be updated by adding the latest references.
  8. Neutrosophic statistics is the extension of classical statistics and is applied when the data is coming from a complex process or from an uncertain environment. The current study can be extended using neutrosophic statistics as future research. The statement that the proposed study can be extended for neutrosophic statistics can be added by citing some papers on neutrosophic statistics. 

Author Response

Sincerely thank you for your precious advice. Please see the attachment to see our revisions.

Round 2

Reviewer 1 Report

After the first revision round, the revised article has improved. The paper is interesting and worthy of publication.

Reviewer 2 Report

The paper can be accepted now